# SteinDreamer: Variance Reduction for Text-to-3D Score Distillation via Stein Identity

## Abstract

Score distillation has emerged as one of the most prevalent approaches for text-to-3D asset synthesis. Essentially, score distillation updates 3D parameters by lifting and back-propagating scores averaged over different views. In this paper, we reveal that the gradient estimation in score distillation is inherent to high variance. Through the lens of variance reduction, the effectiveness of SDS and VSD can be interpreted as applications of various control variates to the Monte Carlo estimator of the distilled score. Motivated by this rethinking and based on Stein's identity, we propose a more general solution to reduce variance for score distillation, termed *Stein Score Distillation (SSD)*. SSD incorporates control variates constructed by Stein identity, allowing for arbitrary baseline functions. This enables us to include flexible guidance priors and network architectures to explicitly optimize for variance reduction. In our experiments, the overall pipeline, dubbed *SteinDreamer*, is implemented by instantiating the control variate with a monocular depth estimator. The results suggest that SSD can effectively reduce the distillation variance and consistently improve visual quality for both object- and scene-level generation. Moreover, we demonstrate that SteinDreamer achieves faster convergence than existing methods due to more stable gradient updates.

## 1 Introduction

There have been recent significant advancements in text-to-image generation, driven by diffusion models. Notable examples include Nichol et al. (2021); Ramesh et al. (2021; 2022); Rombach et al. (2022) and Sohl-Dickstein et al. (2015); Ho et al. (2020); Song & Ermon (2019); Song et al. (2020); Dhariwal & Nichol (2021). These developments have sparked growing interest in the realm of text-guided 3D generation. This emerging field aims to automate and accelerate 3D asset creation in the applications of virtual reality, movies, and gaming. However, 3D synthesis poses significantly greater challenges. Directly training generative models using 3D data, as explored in works by (Wu et al., 2016; Yang et al., 2019; Cai et al., 2020; Nichol et al., 2022; Jun & Nichol, 2023; Chan et al., 2022; Shue et al., 2022), faces practical hurdles due to the scarcity of high-quality and diverse data. Moreover, the inherent complexity of generative modeling with 3D representations adds an extra layer of intricacy to this endeavor.

In recent times, techniques based on score distillation, exemplified by DreamFusion (Poole et al., 2022) and ProlificDreamer (Wang et al., 2023b), have gained prominence. These methods have garnered attention for their ability to effectively bypass the need for 3D data by leveraging a 2D diffusion model for 3D generation. In particular, Poole et al. (2022) introduces Score Distillation Sampling (SDS), which optimizes a differentiable 3D representation, such as NeRF (Mildenhall et al., 2020), by lifting and back-propagating image scores from a pre-trained text-to-image diffusion model. Among its subsequent works (Lin et al., 2023; Wang et al., 2023a; Chen et al., 2023; Metzer et al., 2023), ProlificDreamer (Wang et al., 2023b) stands out for significantly enhancing the generation quality through derived Variational Score Distillation (VSD). VSD introduces an additional score for rendered images to improve parameter updates.

However, it is widely recognized that gradient obtained through score distillation techniques tend to be noisy and unstable due to the high uncertainty in the denoising process and the small batch size limited by computational constraints. Consequently, this leads to slow convergence and suboptimal solutions. In this paper, we address this issue by proposing a unified variance reduction approach.

We reveal that both the noise term in SDS and the extra score function introduced by VSD have zero means, and thus can be regarded as *control variates*. The update of VSD is equivalent to the update of SSD in expectation. However, the gradient variance is smaller in VSD due to the effect of a better implementation the control variate.

Building on these insights, we present a more flexible control variate for score distillation, leveraging Stein identity (Stein, 1972; Chen, 1975; Gorham & Mackey, 2015), dubbed *Stein Score Distillation (SSD)*. Stein's identity, given by $\mathbb{E}_{\boldsymbol{x} \sim p}[\nabla_{\boldsymbol{x}} \log p(\boldsymbol{x}) \cdot f(\boldsymbol{x})^{\top} + \nabla_{\boldsymbol{x}} f(\boldsymbol{x})] = 0$ for any distribution $p$ and function $f$ satisfying mild regularity conditions (Stein, 1972; Liu et al., 2016; Liu, 2017). This formulation establishes a broader class of control variates due to its zero means, providing flexibility in optimizing function $f$ for variance reduction. Specifically, our *Stein Score Distillation (SSD)* frames the distillation update as a combination of the score estimation from a pre-trained diffusion model and a control variate derived from Stein's identity. The first term aligns with with that in SDS and VSD, serving to maximize the likelihood of the rendered image. The second control variate is tailored to specifically reduce gradient variance. Importantly, our construction allows us to incorporate arbitrary prior knowledge and network architectures in $f$, facilitating the design of control variates highly correlated with the lifted image score, leading to a significant reduction in gradient variance.

We integrate our proposed SSD into a text-to-3D generation pipeline, coined as *SteinDreamer*. Through extensive experiments, we demonstrate that SteinDreamer can consistently mitigate variance issues within the score distillation process. For both 3D object and scene-level generation, SteinDreamer outperforms DreamFusion and ProlificDreamer by providing detailed textures, precise geometries, and effective alleviation of the Janus (Hong et al., 2023) and ghostly (Warburg et al., 2023) artifacts. Lastly, it's worth noting that SteinDreamer, with its reduced variance, accelerates the convergence of 3D generation, reducing the number of iterations required by 14%-22%.

## 2 PRELIMINARIES

### 2.1 SCORE DISTILLATION

Diffusion models, as demonstrated by various works (Sohl-Dickstein et al., 2015; Ho et al., 2020; Song & Ermon, 2019; Song et al., 2020), have proven to be highly effective in text-to-image generation. Build upon the success of 2D diffusion models, Poole et al. (2022); Wang et al. (2023a); Lin et al. (2023); Chen et al. (2023); Tsalicoglou et al. (2023); Metzer et al. (2023); Wang et al. (2023b); Huang et al. (2023) demonstrate the feasibility of using a 2D generative model to create 3D asserts. Among these works, score distillation techniques play a central role by providing a way to guide a differentiable 3D representation using a pre-trained text-to-image diffusion model.

Essentially, score distillation lifts and back-propagates signals estimated from a 2D prior to update a differentiable 3D representation, such as NeRF (Mildenhall et al., 2020), via the chain rule (Wang et al., 2023a). There are primarily two types of distillation schemes: Score Distillation Sampling (SDS) (Poole et al., 2022) and Variational Score Distillation (VSD) (Wang et al., 2023b):

**Score Distillation Sampling.** The main idea of SDS is to utilize the denoising score matching loss to optimize a 3D representation that semantically matches a given text prompt $\boldsymbol{y}$ based on its multi-view projection, using the score function of a 2D image distribution $\nabla \log p_t$. By taking derivatives with respect to 3D parameters $\boldsymbol{\theta}$ and dropping the Jacobian matrix of the score function to simplify the computation, SDS yields the following update to optimize $\boldsymbol{\theta}$:

$$\boldsymbol{\Delta}_{SDS} = \mathbb{E}_{t,\boldsymbol{c},\boldsymbol{\epsilon} \sim \mathcal{N}(\boldsymbol{0},\boldsymbol{I})} \left[ \omega(t) \frac{\partial g(\boldsymbol{\theta}, \boldsymbol{c})}{\partial \boldsymbol{\theta}} \left( \sigma_t \nabla \log p_t(\alpha_t g(\boldsymbol{\theta}, \boldsymbol{c}) + \sigma_t \boldsymbol{\epsilon} | \boldsymbol{y}) - \boldsymbol{\epsilon} \right) \right], \quad (1)$$

where the expectation of $t$ is taken over a uniform distribution $\mathcal{U}[0, T]$, and $\alpha_t, \sigma_t > 0$ are time-dependent diffusion coefficients. And $\boldsymbol{c}$ is taken over some camera distribution $p_c$ defined on $\mathbb{SO}(3) \times \mathbb{R}^3$, $g(\boldsymbol{\theta}, \boldsymbol{c})$ renders a 2D view from $\boldsymbol{\theta}$ given $\boldsymbol{c}$. In this work, we follow DreamFusion (Poole et al., 2022) and parameterize $\boldsymbol{\theta}$ as a NeRF. In this case, $g(\boldsymbol{\theta}, \boldsymbol{c})$ renders a view by casting each pixel on the image plane to a ray using camera pose $\boldsymbol{c}$ with volume rendering. Meanwhile, $\nabla \log p_t$ can be surrogated by a noise estimator in a pre-trained diffusion model.

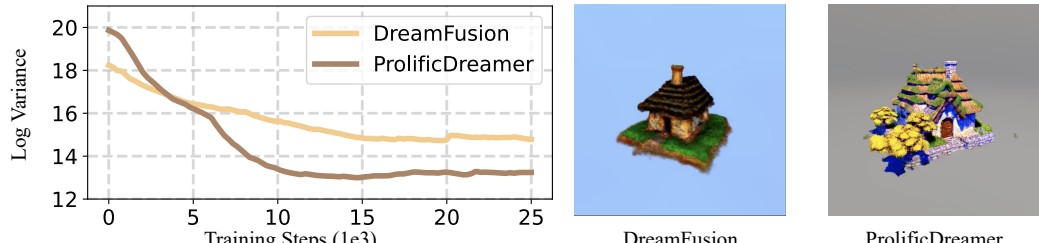

Figure 1: **Variance comparison between SDS and VSD.** We monitor the variance of $\mathbf{\Delta}_{SDS}$ and $\mathbf{\Delta}_{VSD}$ for every 100 training step. We show that variance level is highly correlated to the performance of score distillation.

**Variational Score Distillation.** ProlificDreamer (Wang et al., 2023b) introduced a new variant of score distillation, *variational score distillation (VSD)*, through the lens of particle-based variational inference (Liu & Wang, 2016; Liu, 2017; Detommaso et al., 2018). ProlificDreamer also minimizes the KL divergence between $p_t(\boldsymbol{x})$ and the image distribution rendered from a 3D representation $\boldsymbol{\theta}$. It achieves this by deriving the following update rule through Wasserstein gradient flow:

$$\mathbf{\Delta}_{VSD} = \mathbb{E}_{t,\boldsymbol{c},\boldsymbol{\epsilon}\sim\mathcal{N}(\mathbf{0},\boldsymbol{I})} \left[ \omega(t) \frac{\partial g(\boldsymbol{\theta}, \boldsymbol{c})}{\partial \boldsymbol{\theta}} \left( \sigma_t \nabla \log p_t(\boldsymbol{x}|\boldsymbol{y}) - \sigma_t \nabla \log q_t(\boldsymbol{x}|\boldsymbol{c}) \right) \right], \qquad (2)$$

where $\boldsymbol{x} = \alpha_t g(\boldsymbol{\theta}, \boldsymbol{c}) + \sigma_t \boldsymbol{\epsilon}$ is the noisy observation of the rendered image, sampled by a random camera pose $\boldsymbol{c}$. Notably, there emerges a new score function of probability density function $q_t(\boldsymbol{x}|\boldsymbol{c})$, which characterizes the conditional distribution of noisy rendered images given the camera pose $\boldsymbol{c}$. While $\nabla \log p_t$ can be approximated in a similar manner using an off-the-shelf diffusion model $\boldsymbol{\epsilon}_{\phi^*}$, $\nabla \log q_t$ is not readily available. The solution provided by Wang et al. (2023b) is to fine-tune a pretrained diffusion model using the rendered images. The approach results an alternating optimization paradigm between $\boldsymbol{\theta}$ with Eq. 2 and $\boldsymbol{\psi}$ with score matching:

$$\min_{\boldsymbol{\psi}} \mathbb{E}_{t,\boldsymbol{c},\boldsymbol{\epsilon}\sim\mathcal{N}(\mathbf{0},\boldsymbol{I})} \left[ \omega(t) \| \boldsymbol{\epsilon}_{\boldsymbol{\psi}}(\alpha_t g(\boldsymbol{\theta}, \boldsymbol{c}) + \sigma_t \boldsymbol{\epsilon}, t, \boldsymbol{c}, \boldsymbol{y}) - \boldsymbol{\epsilon} \|_2^2 \right], \qquad (3)$$

where $\boldsymbol{\epsilon}_{\boldsymbol{\psi}}(\boldsymbol{x}, t, \boldsymbol{c}, \boldsymbol{y})$ is a diffusion model additionally conditioned on the camera pose $\boldsymbol{c}$. $\boldsymbol{\psi}$ is initialized with a pre-trained one and parameterized by LoRA (Hu et al., 2021).

### 2.2 CONTROL VARIATE

Meanwhile, in this work, we introduce control variate to the context of score distillation. Control variate is a widely utilized technique to reduce variance for Monte Carlo estimator in various fields, including physical simulation (Davies et al., 2004), graphical rendering (Kajiya, 1986; Müller et al., 2020), network science (Meyn, 2008; Chen et al., 2017), and reinforcement learning (Williams, 1992; Sutton et al., 1998; 1999; Liu et al., 2017). Suppose we want to estimate the expectation $\mathbb{E}_{\boldsymbol{x}\sim q(\boldsymbol{x})}[f(\boldsymbol{x})]$ for some function $f : \mathbb{R}^D \to \mathbb{R}^D$ via Monte Carlo samples $\{\boldsymbol{x}_i \in \mathbb{R}^D\}_{i=1}^N$: $\mathbf{\Delta} = \frac{1}{N} \sum_{i=1}^N f(\boldsymbol{x}_i)$. The estimator $\mathbf{\Delta}$ is supposed to have large variance when $N$ is small. Consider we have control variate as a function $h : \mathbb{R}^D \to \mathbb{R}^D$ with analytic mean under $q(\boldsymbol{x})$, which without loss of generality, can be assumed to have zero mean. Then we can construct an unbiased estimator by adding term $\boldsymbol{\xi} = \frac{1}{N} \sum_{i=1}^N h(\boldsymbol{x}_i)$: $\mathbf{\Delta}^\dagger = \frac{1}{N} \sum_{i=1}^N (f(\boldsymbol{x}_i) + \boldsymbol{\mu} \odot h(\boldsymbol{x}_i))$, where $\boldsymbol{u} \in \mathbb{R}^D$ is a group of reweighting coefficients and $\odot$ denotes element-wise multiplication. The resultant estimator has variance for the $i$-th entry:

$$\text{Var}\left[\mathbf{\Delta}_i^\dagger\right] = \text{Var}\left[\mathbf{\Delta}_i\right] + \boldsymbol{\mu}_i^2 \text{Var}\left[\boldsymbol{\xi}_i\right] + 2\boldsymbol{\mu}_i \mathbb{E}\left[\mathbf{\Delta}\boldsymbol{\xi}^\top\right]_{ii}, \qquad (4)$$

where it is possible to reduce $\text{Var}[\mathbf{\Delta}_i^\dagger]$ by selecting $h$ and $\boldsymbol{u}$ properly. To maximize variance reduction, $\boldsymbol{u}$ is chosen as $\boldsymbol{u}_i^* = -\mathbb{E}[\mathbf{\Delta}\boldsymbol{\xi}^\top]_{ii}/\text{Var}[\boldsymbol{\xi}_i]$, leading to $\text{Var}[\mathbf{\Delta}_i^\dagger] = (1 - \text{Corr}(\mathbf{\Delta}_i, \boldsymbol{\xi}_i)^2)\text{Var}[\mathbf{\Delta}_i]$, where $\text{Corr}(\cdot, \cdot)$ denotes the correlation coefficient. This signifies that higher correlation between functions $f$ and $h$, then more variance can be reduced.

## 3 RETHINKING SDS AND VSD: A CONTROL VARIATE PERSPECTIVE

In this section, we reveal that the variance of update estimation may play a key role in score distillation. At first glance, SDS and VSD differ in their formulation and implementation. However, our first theoretical finding reveals that SDS and (single-particle) VSD are equivalent in their expectation, i.e., $\Delta_{SDS} = \Delta_{VSD}$. We formally illustrate this observation below.

As a warm-up, we inspect SDS via the following rewriting.

$$\Delta_{SDS} = \mathbb{E}_{t,\boldsymbol{c},\boldsymbol{\epsilon}} \underbrace{\left[ \omega(t) \frac{\partial g(\boldsymbol{\theta}, \boldsymbol{c})}{\partial \boldsymbol{\theta}} \sigma_t \nabla \log p_t(\boldsymbol{x}|\boldsymbol{y}) \right]}_{f(t, \boldsymbol{\theta}, \boldsymbol{x}, \boldsymbol{c})} - \mathbb{E}_{t,\boldsymbol{c},\boldsymbol{\epsilon}} \underbrace{\left[ \omega(t) \frac{\partial g(\boldsymbol{\theta}, \boldsymbol{c})}{\partial \boldsymbol{\theta}} \boldsymbol{\epsilon} \right]}_{h_{SDS}(t, \boldsymbol{\theta}, \boldsymbol{x}, \boldsymbol{c})}, \tag{5}$$

where $\boldsymbol{x} = \alpha_t g(\boldsymbol{\theta}, \boldsymbol{c}) + \sigma_t \boldsymbol{\epsilon}$. The second term $\mathbb{E}[h_{SDS}(t, \boldsymbol{\theta}, \boldsymbol{x}, \boldsymbol{c})] = 0$ simply because it is the expectation of a zero-mean Gaussian vector. For VSD, we follow a similar derivation and obtain:

$$\Delta_{VSD} = \mathbb{E}_{t,\boldsymbol{c},\boldsymbol{\epsilon}} \underbrace{\left[ \omega(t) \frac{\partial g(\boldsymbol{\theta}, \boldsymbol{c})}{\partial \boldsymbol{\theta}} \sigma_t \nabla \log p_t(\boldsymbol{x}|\boldsymbol{y}) \right]}_{f(t, \boldsymbol{\theta}, \boldsymbol{x}, \boldsymbol{c})} - \mathbb{E}_{t,\boldsymbol{c},\boldsymbol{\epsilon}} \underbrace{\left[ \omega(t) \frac{\partial g(\boldsymbol{\theta}, \boldsymbol{c})}{\partial \boldsymbol{\theta}} \sigma_t \nabla \log q_t(\boldsymbol{x}|\boldsymbol{c}) \right]}_{h_{VSD}(t, \boldsymbol{\theta}, \boldsymbol{x}, \boldsymbol{c})}, \tag{6}$$

where once again the second term $\mathbb{E}[h_{VSD}(t, \boldsymbol{\theta}, \boldsymbol{x}, \boldsymbol{c})] = 0$. This can be proven by showing that $q_t(\boldsymbol{x}|\boldsymbol{c})$ turns out to be a zero-mean Gaussian distribution or applying the inverse chain rule followed by the fact that the first-order moment of the score function is constantly zero. Moreover, the first term $\mathbb{E}[f(t, \boldsymbol{\theta}, \boldsymbol{x}, \boldsymbol{c})]$ of both SDS and VSD equals to $-\nabla_{\boldsymbol{\theta}} \mathbb{E}_t [D_{\mathrm{KL}}(q_t(\boldsymbol{x}|\boldsymbol{c}) \| p_t(\boldsymbol{x}|\boldsymbol{y}))]$. This implies that SDS and VSD are equivalently minimizing the distribution discrepancy between the noisy rendered image distribution and the Gaussian perturbed true image distribution, as a gradient descent algorithm. We defer the full derivation to Appendix A.

However, in most scenarios, empirical evidence indicates that VSD consistently outperforms SDS, despite both methods aiming to minimize the same underlying objective. To explain this paradox, we posit that the underlying source of their performance disparities is attributed to the *variance* of stochastic simulation of the expected updates suggested by SDS and VSD. The numerical evaluation of Eq. 1 and Eq. 2 typically relies on Monte Carlo estimation over a mini-batch. Unfortunately, rendering a full view from NeRF and performing inference with diffusion models are computationally demanding processes, leading to constraints on the number of rendered views that can be utilized within a single optimization step, often limited to just one, as observed in previous work (Poole et al., 2022). Additionally, the term related to the score function within the expectation undergoes a denoising procedure, notorious for its instability and high uncertainty, especially when $t$ is large. Hence, despite SDS and VSD having identical means, we argue that the variance of their numerical estimation significantly differs. We empirically validate this hypothesis in Fig. 1, where we visualize the variance of $\Delta_{SDS}$ and $\Delta_{VSD}$ during the training process. We observe that VSD yields results with richer textures, and in the meanwhile, achieves lower level variance compared to SDS.

To gain insight into the variance disparity between SDS and VSD, we connect SDS and VSD via the concept of control variates. As introduced in Sec. 2.2, a control variate is a zero-mean random variable capable of reducing the variance of Monte Carlo estimator when incorporated into the simulated examples. Notably, both $h_{SDS}(t, \boldsymbol{\theta}, \boldsymbol{x}, \boldsymbol{c})$ and $h_{VSD}(t, \boldsymbol{\theta}, \boldsymbol{x}, \boldsymbol{c})$ can be regarded as control variates, as confirmed by Eq. 5 and Eq. 6 due to their zero means. Consequently, SDS and VSD can be interpreted as Monte Carlo estimators of the gradient of the KL divergence, integrated with different control variates. As demonstrated in Sec. 2.2, control variate with higher correlation to the estimated variable leads to larger variance reduction. VSD exhibits lower variance primarily because $\nabla_{\boldsymbol{x}} \log q_t(\boldsymbol{x}|\boldsymbol{c})$ in control variate $h_{VSD}$ is fine-tuned from $\nabla_{\boldsymbol{x}} \log p_t(\boldsymbol{x}|\boldsymbol{c})$, and thus resulting in higher correlation compared to the pure Gaussian noises in $h_{SDS}$.

## 4 STEIN SCORE DISTILLATION

Having revealed that variance control is one of the key knobs to improve the performance of score distillation, we extend the family of control variates that can be used for score distillation in this section.

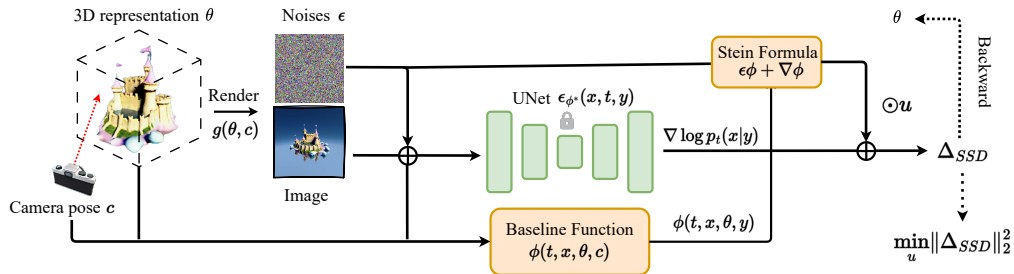

Figure 2: **Pipeline of SteinDreamer.** We incorporate control variates constructed by Stein's identity into a score distillation pipeline, allowing for arbitrary baseline functions. In practice, we implement the baseline functions with a monocular depth estimator or normal estimator.

## 4.1 STEIN CONTROL VARIATES FOR SCORE DISTILLATION

Our main inspiration is drawn from Oates et al. (2017); Liu (2017); Roeder et al. (2017) that Stein's identity can be served as a powerful and flexible tool to construct zero-mean random variables. We consider Stein's identity associated with any conditional probability $p(\boldsymbol{x}|\boldsymbol{\theta}, \boldsymbol{c})$ as below:

$$\mathbb{E}_{\boldsymbol{x} \sim p(\boldsymbol{x}|\boldsymbol{\theta}, \boldsymbol{c})} \left[ \nabla \log p(\boldsymbol{x}|\boldsymbol{\theta}, \boldsymbol{c}) \phi(t, \boldsymbol{\theta}, \boldsymbol{x}, \boldsymbol{c}) + \nabla_{\boldsymbol{x}} \phi(t, \boldsymbol{\theta}, \boldsymbol{x}, \boldsymbol{c}) \right] = \boldsymbol{0}, \tag{7}$$

where $\phi(t, \boldsymbol{\theta}, \boldsymbol{x}, \boldsymbol{c})$ is referred to as the *baseline function*, which can be arbitrary scalar-value function satisfying regularity conditions (Stein, 1972; Liu et al., 2016; Liu, 2017). By plugging $q_t(\boldsymbol{x}|\boldsymbol{\theta}, \boldsymbol{c})$ into Eq. 7, we can construct our control variate as follows:

$$h_{SSD}(t, \boldsymbol{\theta}, \boldsymbol{c}, \boldsymbol{x}) = \omega(t) \frac{\partial g(\boldsymbol{\theta}, \boldsymbol{c})}{\partial \boldsymbol{\theta}} \left( \boldsymbol{\epsilon} \phi(t, \boldsymbol{\theta}, \boldsymbol{x}, \boldsymbol{c}) + \nabla_{\boldsymbol{x}} \phi(t, \boldsymbol{\theta}, \boldsymbol{x}, \boldsymbol{c}) \right), \tag{8}$$

where $\boldsymbol{x} = \alpha_t g(\boldsymbol{\theta}, \boldsymbol{c}) + \sigma_t \boldsymbol{\epsilon}$ and $\boldsymbol{\epsilon} \sim \mathcal{N}(\boldsymbol{0}, \boldsymbol{I})$. Additional details and derivations are provided in Appendix A. The advantage of $h_{SSD}$ lies in its flexibility to define an infinite class of control variates, characterized by arbitrary baseline function $\phi(t, \boldsymbol{\theta}, \boldsymbol{x}, \boldsymbol{c})$.

## 4.2 VARIANCE MINIMIZATION VIA STEIN SCORE DISTILLATION

We propose to adopt $h_{SSD}$ as the control variate for score distillation. In addition to $h_{SSD}$, we introduce a group of learnable weights $\boldsymbol{\mu} \in \mathbb{R}^D$ to facilitate optimal variance reduction following the standard scheme introduced in Sec. 2.2. Altogether, we present the following update rule, termed as *Stein Score Distillation (SSD)*:

$$\boldsymbol{\Delta}_{SSD} = \mathbb{E}_{t, \boldsymbol{c}, \boldsymbol{\epsilon}} \left[ \omega(t) \frac{\partial g(\boldsymbol{\theta}, \boldsymbol{c})}{\partial \boldsymbol{\theta}} \left( \sigma_t \nabla \log p_t(\boldsymbol{x}|\boldsymbol{y}) + \boldsymbol{\mu} \odot [\boldsymbol{\epsilon} \phi(t, \boldsymbol{\theta}, \boldsymbol{x}, \boldsymbol{c}) + \nabla_{\boldsymbol{x}} \phi(t, \boldsymbol{\theta}, \boldsymbol{x}, \boldsymbol{c})] \right) \right]. \tag{9}$$

Here $\phi(t, \boldsymbol{\theta}, \boldsymbol{x}, \boldsymbol{c})$ can be instantiated using any neural network architecture taking 3D parameters, noisy rendered image, and camera pose as the input.

In our experiments, we employ a pre-trained monocular depth estimator, MiDAS (Ranftl et al., 2020; 2021), coupled with domain-specific loss functions to construct $\phi(t, \boldsymbol{\theta}, \boldsymbol{x}, \boldsymbol{c})$, as a handy yet effective choice. Specifically:

$$\phi(t, \boldsymbol{x}, \boldsymbol{\theta}, \boldsymbol{c}) = -\ell(\alpha(\boldsymbol{\theta}, \boldsymbol{c}), \text{MiDAS}(\boldsymbol{x})). \tag{10}$$

Here $\text{MiDAS}(\cdot)$ can estimate either depth or normal map from noisy observation $\boldsymbol{x}$. And $\alpha(\cdot, \cdot)$ is chosen as the corresponding depth or normal renderer of the 3D representation $\boldsymbol{\theta}$, and $\ell(\cdot, \cdot)$ is the Pearson correlation loss when estimating depth map or cosine similarity loss when considering normal map.

As introduced in Sec. 2.2, there exists a closed-form $\boldsymbol{\mu}$ that maximizes the variance reduction. However, it assumes the correlation between the control variate and the random variable of interest is

known. Instead, we propose to directly optimize variance by adjusting $\boldsymbol{\mu}$ to minimize the second-order moment of Eq. 9 since its first-order moment is independent of $\boldsymbol{\mu}$:

$$\min_{\boldsymbol{\mu}} \mathbb{E}_{t,\boldsymbol{c},\boldsymbol{\epsilon}} \left[ \left\| \omega(t) \frac{\partial g(\boldsymbol{\theta}, \boldsymbol{c})}{\partial \boldsymbol{\theta}} \left( \sigma_t \nabla \log p_t(\boldsymbol{x}|\boldsymbol{y}) + \boldsymbol{\mu} \odot [\boldsymbol{\epsilon}\phi(t, \boldsymbol{\theta}, \boldsymbol{x}, \boldsymbol{c}) + \nabla_{\boldsymbol{x}}\phi(t, \boldsymbol{\theta}, \boldsymbol{x}, \boldsymbol{c})] \right) \right\|_2^2 \right], \quad (11)$$

which essentially imposes a penalty on the gradient norm of $\boldsymbol{\theta}$. We alternate between optimizing $\boldsymbol{\theta}$ and $\boldsymbol{\mu}$ using SSD gradient in Eq. 9 and the objective function in Eq. 11, respectively. We refer to our complete text-to-3D framework as *SteinDreamer*, and its optimization paradigm is illustrated in Fig. 2.

Specifically, during each optimization iteration, SteinDreamer performs the following steps: 1) renders RGB map and depth/normal map from a random view of $\boldsymbol{\theta}$, 2) perturbs the RGB map and obtains the score estimation using a pre-trained diffusion model and monocular depth/normal prediction from a pre-trained MiDAS, 3) computes $\phi$ via Eq. 10 and its gradient via auto-differentiation to form control variate $h_{SSD}$, 4) weights the control variate by $\boldsymbol{\mu}$ and combine it with the diffusion score $\nabla \log p_t(\boldsymbol{x}|\boldsymbol{y})$, 5) back-propagates $\boldsymbol{\Delta}_{SSD}$ through the chain rule to update 3D parameters $\boldsymbol{\theta}$. In the other fold, SteinDreamer keeps $\boldsymbol{\theta}$ frozen and optimizes $\boldsymbol{\mu}$ to minimize the $\ell_2$ norm of the update signals on $\boldsymbol{\theta}$ according to Eq. 11.

### 4.3 DISCUSSION

In this section, we discuss a few merits of our *Stein score distillation* (SSD) in comparison to SDS and VSD. First, SDS is a special case of SSD when taking $\phi(t, \boldsymbol{\theta}, \boldsymbol{x}, \boldsymbol{c}) = -1$. This observation suggests the potential for SSD to provide a lower variance in gradient estimation due to its broader range in representing control variates. As demonstrated in Oates et al. (2017), an optimal control variate can be constructed using Stein's identity by carefully selecting $\phi$, achieving a zero-variance estimator. The key advantage of SSD lies in its flexibility in choosing the baseline function $\phi$, which can directly condition and operate on all relevant variables. Furthermore, the expressive power of SSD surpasses that of VSD, in which $\nabla \log q_t(\boldsymbol{x}|\boldsymbol{c})$ implicitly conditions on $\boldsymbol{\theta}$ through $\boldsymbol{x}$ and $\boldsymbol{c}$.

Kim et al. (2023) proposes Collaborative Score Distillation (CSD) to sample latent parameters via Stein Variational Gradient Descent (SVGD). While both methods are grounded in Stein's method, the underlying principles significantly differ. In CSD, the SVGD-based update takes the form of the Stein discrepancy: $\max_{\phi \in \mathcal{F}} \mathbb{E}_{\boldsymbol{x} \sim q(\boldsymbol{x})}[\phi(\boldsymbol{x})\nabla \log p(\boldsymbol{x}) + \nabla\phi(\boldsymbol{x})]$, where $\phi(\boldsymbol{x})$ is often interpreted as an update direction constrained by a function class $\mathcal{F}$ (RBF kernel space in (Kim et al., 2023)). In contrast, our update rule appends a zero-mean random variable via the Stein identity after the raw gradient of the KL divergence (Eq. 9), where $\phi(x)$ typically represents a pre-defined baseline function. The potential rationale behind CSD to reducing variance lies in introducing the RBF kernel as a prior to constrain the solution space by modeling pairwise relations between data samples. Our SSD is centered around constructing a more general control variate that correlates with the random variable of interest, featuring zero mean but variance reduction.

## 5 EXPERIMENTS

We conduct experiments for both object-level and scene-level text-to-3d generation. The text prompts utilized in the experiments are originally from ProlificDreamer (Wang et al., 2023b). We mainly compare against the seminal works DreamFusion (Poole et al., 2022) and ProlificDreamer (Wang et al., 2023b). For a fair comparison, we utilize the open-source threestudio [1] as a unified benchmarking implementation. We thoroughly test our proposed SteinDreamer with both depth estimator and normal estimator priors. All training hyper-parameters are kept the same with ProlificDreamer. For simplicity, we evaluate ProlificDreamer with the particle number set to one.

### 5.1 RESULTS AND ANALYSIS

**Object Centric Generation.** We put our qualitative results in Fig. 3a and Fig. 3b for Stein-Dreamer with Depth or normal prior, respectively. Compared with SDS loss from DreamFusion, our SteinDreamer presents novel views with less over-saturation and over-smoothing artifacts. When

---

[1] https://github.com/threestudio-project/threestudio

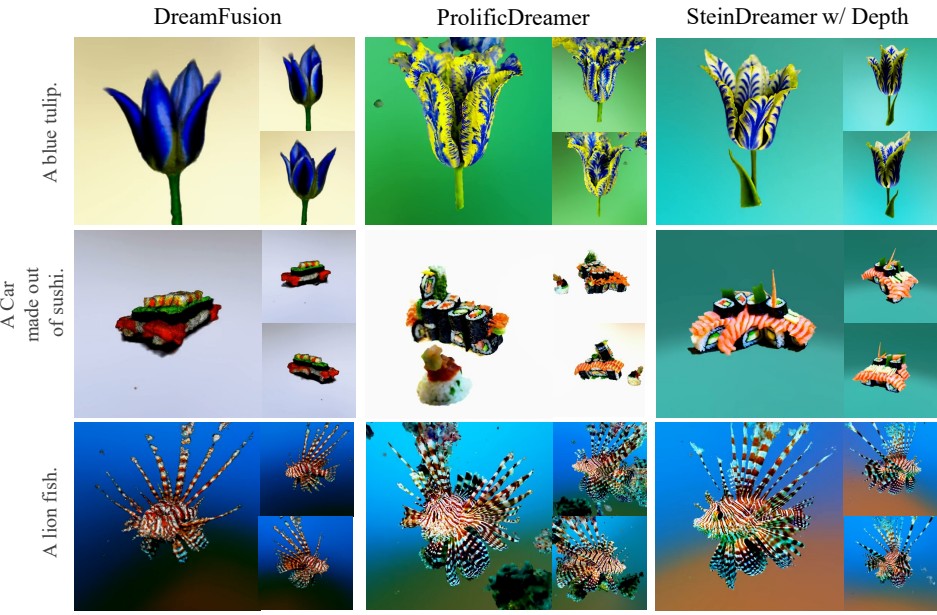

(a) Qualitative comparisons between SteinDreamer w/ depth estimator and existing methods.

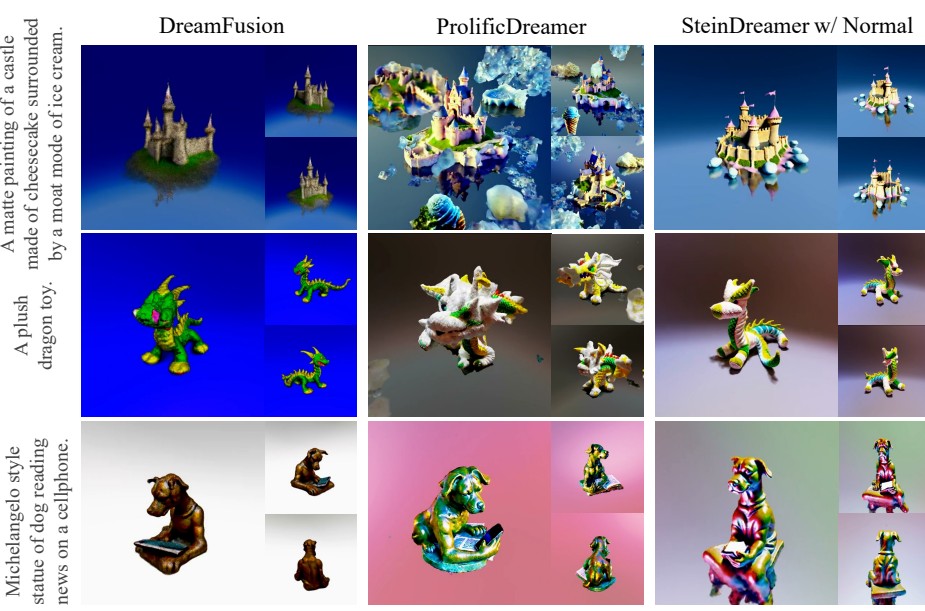

(b) Qualitative comparisons between SteinDreamer w/ normal estimator and existing methods.

Figure 3: **Object-level comparisons.** Compared to existing methods, our SteinDreamer delivers smoother geometry, more detailed texture, and fewer floater artifacts.

comparing with VSD loss from ProlificDreamer, not only does our SteinDreamer generate smoother geometry, but also delivers sharper textures without contamination of floaters. Additionally, it's worth noting that our SteinDreamer also alleviates the Janus problem. As shown in the dog statue case, there is only one face produced in our SteinDreamer's output, while previous methods suffer from the multi-face issue. We further monitor the variance for all the demonstrated examples during the training stage in Fig. 5. Variance is estimated by 120 randomly sampled tuples of time steps, camera poses, and Gaussian noises. It is clear that our SteinDreamer consistently has lower variance than compared baselines throughout the course of training. Quantitative comparison can be found in Appendix C.1.

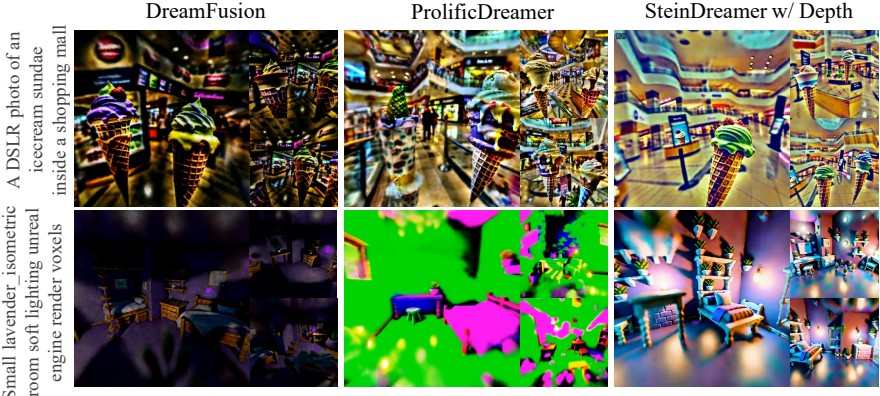

(a) Qualitative comparisons between SteinDreamer w/ depth estimator and existing methods.

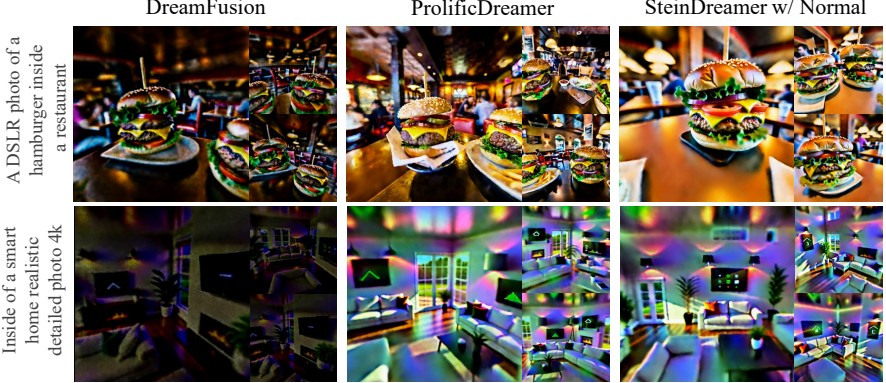

(b) Qualitative comparisons between SteinDreamer w/ normal estimator and existing methods.

Figure 4: **Scene-level comparisons between DreamFusion, ProlificDreamer, and SteinDreamer.** Compared to existing methods, SteinDreamer presents more realistic textures with better details.

**Large Scene Generation.** We further investigate the performance of our method and the comparison baselines on a more challenging scenario for scene generation. We provide detailed comparisons for 360°scene-level generation in Fig. 4a and Fig. 4b. DreamFusion delivers blurry results with unrealistic colors and textures. ProlificDreamer's results suffer from the noisy background, and we also observe that the VSD loss can diverge in the texture refining process (Fig. 4a ). In comparison, we observe that results generated by SteinDreamer are much sharper in appearance and enjoy better details. More results are deferred to Appendix C.4.

## 5.2 ABLATION STUDIES

To further validate the effectiveness of our proposed components, we conduct ablation studies on whether or not to employ Eq. 11 to minimize second-order moment. The alternative candidate is to fix $\mu$ as all-one vector during training. As shown in Fig. 6, when explicitly minimizing the variance, we reach cleaner results with better high-frequency signals. The results when optimizing without variance minimization, on the other hand, turned out to generate blurry geometry and noisy textures. It is worth mentioning that excessive variance reduction may smoothen out some necessary details, especially in the background regions, as the left-hand side result contains more detailed background textures than the right-hand side one.

## 5.3 CONVERGENCE SPEED

We also study the convergence speed of our methods as well as compared baselines. Specifically, we use the average CLIP distance (Xu et al., 2022) between the rendered images and the input text prompts as the quality metric. During the training process, we render the 3D assets into multi-view

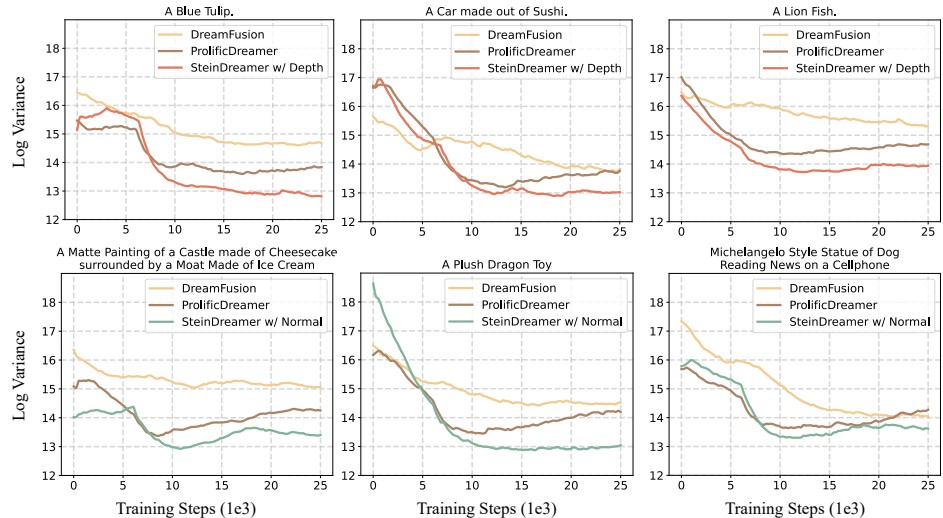

Figure 5: **Variance comparison of $\Delta_{SDS}$, $\Delta_{VSD}$, and $\Delta_{SSD}$.** We visualize how the variance of the investigated three methods for every 1,000 steps. The variance decays as the training converges while $\Delta_{SSD}$ consistently achieves lower variance throughout the whole process.

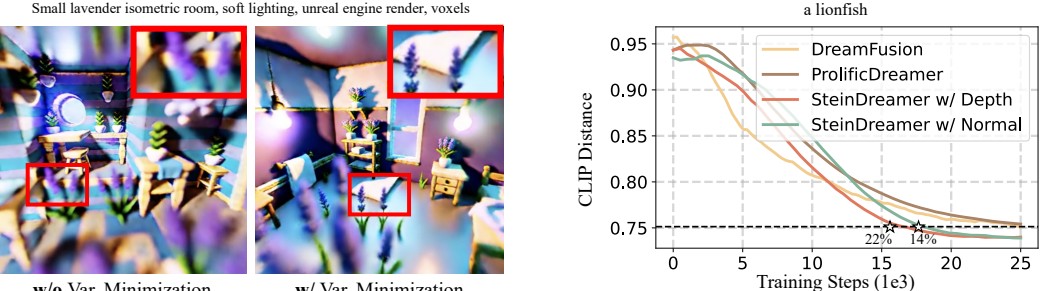

Figure 6: **Ablation study on explicit variance minimization.** We study the effect of turning on/off the optimization step for $\mu$ with respect to loss Eq. 11.

Figure 7: **Convergence speed comparison.** With the help of more stable gradient updates, SteinDreamer accelerates the training process by 14%-22%.

images every 1,000 training steps. In each training step, the diffusion model is inference twice through the classifier-free guidance, which is the same protocol in all compared methods. In Fig. 7, we profile the training steps needed for each approach to reach 0.75 CLIP distance as a desirable threshold. We observe that the proposed SteinDreamer can effectively attain rapid and superior convergence, saving 14%-22% calls of diffusion models. This means that lower variance in our distillation process can speed up convergence. Our SteinDreamer utilizes fewer number of score function evaluations to achieve distilled 3D results that are more aligned with the text prompts. Moreover, since SteinDreamer avoids inferencing and fine-tuning another diffusion model, each iteration of SSD is approximately 2.5-3 times faster than VSD.

## 6 CONCLUSIONS

In this work, we present SteinDreamer, revealing a more general solution to reduce variance for score distillation. Our Stein Score Distillation (SSD) incorporates control variates through Stein identity, admitting arbitrary baseline functions conditioned on all relevant variables with any guidance priors. The experimental results suggest that SSD can effectively reduce the distillation variance and consistently improve visual quality for both object- and scene-level generations. We also showcase that SSD achieves faster and better convergence than existing methods with the help of more stable gradient updates.

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

# A    DEFERRED PROOFS

**Gradient of KL divergence.**    Let $\boldsymbol{\theta}$ parameterize the underlying 3D representation, such as mesh, point clouds, triplanes, or NeRF. We intend to optimize $\boldsymbol{\theta}$ such that each view matches the prior of 2D distribution. This can be formulated by minimizing the KL divergence below:

$$\min_{\boldsymbol{\theta}} \mathbb{E}_{t,\boldsymbol{c}\sim p(\boldsymbol{c})} D_{\mathrm{KL}}(q_t(\boldsymbol{x}|\boldsymbol{\theta},\boldsymbol{c})\|p_t(\boldsymbol{x}|\boldsymbol{y})), \tag{12}$$

where $\boldsymbol{c}$ is the camera pose sampled from a prior distribution, $\boldsymbol{y}$ is the user-specified text prompt, and $q_t(\boldsymbol{x}|\boldsymbol{\theta},\boldsymbol{c}) = \mathcal{N}(\boldsymbol{x}|\alpha_t g(\boldsymbol{\theta},\boldsymbol{c}), \sigma_t^2\boldsymbol{I})$, where $g(\boldsymbol{\theta},\boldsymbol{c})$ is a differentiable renderer that displays scene $\boldsymbol{\theta}$ from the camera angle $\boldsymbol{c}$.

To optimize Eq. 12, we take the gradient in terms of $\boldsymbol{\theta}$ and derive the following update formula:

$$\nabla_{\boldsymbol{\theta}} \mathbb{E}_{t,\boldsymbol{c}} D_{\mathrm{KL}}(q_t(\boldsymbol{x}|\boldsymbol{\theta},\boldsymbol{c})\|p_t(\boldsymbol{x}|\boldsymbol{y})) = \mathbb{E}_{t,\boldsymbol{c}} \nabla_{\boldsymbol{\theta}} D_{\mathrm{KL}}(q_t(\boldsymbol{x}|\boldsymbol{\theta},\boldsymbol{c})\|p_t(\boldsymbol{x}|\boldsymbol{y})) \tag{13}$$

$$= \mathbb{E}_{t,\boldsymbol{c}} \nabla_{\boldsymbol{\theta}} \mathbb{E}_{\boldsymbol{x}\sim q_t(\boldsymbol{x}|\boldsymbol{\theta},\boldsymbol{c})} \left[ \log \frac{q_t(\boldsymbol{x}|\boldsymbol{\theta},\boldsymbol{c})}{p_t(\boldsymbol{x}|\boldsymbol{y})} \right] \tag{14}$$

$$= \mathbb{E}_{t,\boldsymbol{c},\boldsymbol{\epsilon}\sim\mathcal{N}(\boldsymbol{0},\sigma_t^2\boldsymbol{I})} \left[ \underbrace{\nabla_{\boldsymbol{\theta}} \log q_t(\alpha_t g(\boldsymbol{\theta},\boldsymbol{c}) + \boldsymbol{\epsilon}|\boldsymbol{\theta},\boldsymbol{c})}_{(a)} - \underbrace{\nabla_{\boldsymbol{\theta}} \log p_t(\alpha_t g(\boldsymbol{\theta},\boldsymbol{c}) + \boldsymbol{\epsilon}|\boldsymbol{y})}_{(b)} \right] \tag{15}$$

We notice that $q_t(\alpha_t g(\boldsymbol{\theta},\boldsymbol{c}) + \boldsymbol{\epsilon}|\boldsymbol{\theta},\boldsymbol{c}) = \mathcal{N}(\boldsymbol{\epsilon}|\boldsymbol{0}, \sigma_t^2\boldsymbol{I})$, which is independent of $\boldsymbol{\theta}$. Thus $(a) = \boldsymbol{0}$. For term $(b)$, we have:

$$\nabla_{\boldsymbol{\theta}} \log p_t(\alpha_t g(\boldsymbol{\theta},\boldsymbol{c}) + \boldsymbol{\epsilon}|\boldsymbol{y}) = \alpha_t \frac{\partial g(\boldsymbol{\theta},\boldsymbol{c})}{\partial \boldsymbol{\theta}} \nabla \log p_t(\alpha_t g(\boldsymbol{\theta},\boldsymbol{c}) + \boldsymbol{\epsilon}|\boldsymbol{y}). \tag{16}$$

Therefore, $\boldsymbol{\theta}$ should be iteratively updated by:

$$\mathbb{E}_{\boldsymbol{c},\boldsymbol{\epsilon}} \left[ \alpha_t \frac{\partial g(\boldsymbol{\theta},\boldsymbol{c})}{\partial \boldsymbol{\theta}} \nabla \log p_t(\alpha_t g(\boldsymbol{\theta},\boldsymbol{c}) + \boldsymbol{\epsilon}|\boldsymbol{y}) \right] = \mathbb{E}_{\boldsymbol{c},\boldsymbol{x}\sim q_t(\boldsymbol{x}|\boldsymbol{\theta},\boldsymbol{c})} \left[ \alpha_t \frac{\partial g(\boldsymbol{\theta},\boldsymbol{c})}{\partial \boldsymbol{\theta}} \nabla \log p_t(\boldsymbol{x}|\boldsymbol{y}) \right] \tag{17}$$

**SDS equals to the gradient of KL.**    By the following derivation, we demonstrate that SDS essentially minimizes the KL divergence: $\boldsymbol{\Delta}_{SDS} = \nabla_{\boldsymbol{\theta}} \mathbb{E}_{t,\boldsymbol{c}} D_{\mathrm{KL}}(q_t(\boldsymbol{x}|\boldsymbol{\theta},\boldsymbol{c})\|p_t(\boldsymbol{x}|\boldsymbol{y}))$:

$$\mathbb{E}_{\boldsymbol{c},\boldsymbol{x}\sim q_t(\boldsymbol{x}|\boldsymbol{\theta},\boldsymbol{c})} \left[ \frac{\partial g(\boldsymbol{\theta},\boldsymbol{c})}{\partial \boldsymbol{\theta}} (\nabla \log p_t(\boldsymbol{x}|\boldsymbol{y}) - \boldsymbol{\epsilon}) \right] \tag{18}$$

$$= \mathbb{E}_{\boldsymbol{c},\boldsymbol{x}\sim q_t(\boldsymbol{x}|\boldsymbol{\theta},\boldsymbol{c})} \left[ \alpha_t \frac{\partial g(\boldsymbol{\theta},\boldsymbol{c})}{\partial \boldsymbol{\theta}} \nabla \log p_t(\boldsymbol{x}|\boldsymbol{y}) \right] - \underbrace{\mathbb{E}_{\boldsymbol{c},\boldsymbol{\epsilon}\sim\mathcal{N}(\boldsymbol{0},\sigma_t^2\boldsymbol{I})} \left[ \alpha_t \frac{\partial g(\boldsymbol{\theta},\boldsymbol{c})}{\partial \boldsymbol{\theta}} \boldsymbol{\epsilon} \right]}_{=\boldsymbol{0}}. \tag{19}$$

**VSD equals to the gradient of KL.**    We show that VSD also equals to the gradient of KL $\boldsymbol{\Delta}_{VSD} = \nabla_{\boldsymbol{\theta}} \mathbb{E}_{t,\boldsymbol{c}} D_{\mathrm{KL}}(q_t(\boldsymbol{x}|\boldsymbol{\theta},\boldsymbol{c})\|p_t(\boldsymbol{x}|\boldsymbol{y}))$ due to the simple fact that the first-order of score equals to zero:

$$\mathbb{E}_{\boldsymbol{c},\boldsymbol{x}\sim q_t(\boldsymbol{x}|\boldsymbol{\theta},\boldsymbol{c})} \left[ \alpha_t \frac{\partial g(\boldsymbol{\theta},\boldsymbol{c})}{\partial \boldsymbol{\theta}} \nabla \log q_t(\boldsymbol{x}|\boldsymbol{\theta},\boldsymbol{c}) \right] = \mathbb{E}_{\boldsymbol{c},\boldsymbol{x}\sim q_t(\boldsymbol{x}|\boldsymbol{\theta},\boldsymbol{c})} \left[ \nabla_{\boldsymbol{\theta}} \log q_t(\boldsymbol{x}|\boldsymbol{\theta},\boldsymbol{c}) \right] \tag{20}$$

$$= \mathbb{E}_{\boldsymbol{c}} \left[ \int \frac{\nabla_{\boldsymbol{\theta}} q_t(\boldsymbol{x}|\boldsymbol{\theta},\boldsymbol{c})}{q_t(\boldsymbol{x}|\boldsymbol{\theta},\boldsymbol{c})} q_t(\boldsymbol{x}|\boldsymbol{\theta},\boldsymbol{c}) d\boldsymbol{x} \right] \tag{21}$$

$$= \mathbb{E}_{\boldsymbol{c}} \left[ \nabla_{\boldsymbol{\theta}} \int q_t(\boldsymbol{x}|\boldsymbol{\theta},\boldsymbol{c}) d\boldsymbol{x} \right] = \boldsymbol{0}. \tag{22}$$

**Control Variate for SSD.**    Due to Stein's identity, the following is constantly zero:

$$\mathbb{E}_{\boldsymbol{x}\sim q_t(\boldsymbol{x}|\boldsymbol{\theta},\boldsymbol{c})} \left[ \nabla \log q_t(\boldsymbol{x}|\boldsymbol{\theta},\boldsymbol{c}) \phi(t,\boldsymbol{\theta},\boldsymbol{x},\boldsymbol{c}) + \nabla_{\boldsymbol{x}} \phi(t,\boldsymbol{\theta},\boldsymbol{x},\boldsymbol{c}) \right] = \boldsymbol{0}. \tag{23}$$

Plug into Eq. 17, we can obtain:

$$\mathbb{E}_{\boldsymbol{c},\boldsymbol{x}\sim q_t(\boldsymbol{x}|\boldsymbol{\theta},\boldsymbol{c})}\left[\alpha_t\frac{\partial g(\boldsymbol{\theta},\boldsymbol{c})}{\partial\boldsymbol{\theta}}\nabla\log p_t(\boldsymbol{x}|\boldsymbol{y})\right] \tag{24}$$

$$=\mathbb{E}_{\boldsymbol{c}}\left[\omega(t)\frac{\partial g(\boldsymbol{\theta},\boldsymbol{c})}{\partial\boldsymbol{\theta}}\mathbb{E}_{\boldsymbol{x}\sim q_t(\boldsymbol{x}|\boldsymbol{\theta},\boldsymbol{c})}\left[\nabla\log p_t(\boldsymbol{x}|\boldsymbol{y})+\nabla\log q_t(\boldsymbol{x}|\boldsymbol{\theta},\boldsymbol{c})\phi(t,\boldsymbol{\theta},\boldsymbol{x},\boldsymbol{c})+\nabla_{\boldsymbol{x}}\phi(t,\boldsymbol{\theta},\boldsymbol{x},\boldsymbol{c})\right]\right] \tag{25}$$

$$=\mathbb{E}_{\boldsymbol{c}}\left[\omega(t)\frac{\partial g(\boldsymbol{\theta},\boldsymbol{c})}{\partial\boldsymbol{\theta}}\mathbb{E}_{\boldsymbol{x}\sim q_t(\boldsymbol{x}|\boldsymbol{\theta},\boldsymbol{c})}\left[\nabla\log p_t(\boldsymbol{x}|\boldsymbol{y})+\boldsymbol{\epsilon}\phi(t,\boldsymbol{\theta},\boldsymbol{x},\boldsymbol{c})+\nabla_{\boldsymbol{x}}\phi(t,\boldsymbol{\theta},\boldsymbol{x},\boldsymbol{c})\right]\right] \tag{26}$$

$$=\mathbb{E}_{\boldsymbol{c},\boldsymbol{\epsilon}}\left[\omega(t)\frac{\partial g(\boldsymbol{\theta},\boldsymbol{c})}{\partial\boldsymbol{\theta}}\left(\nabla\log p_t(\boldsymbol{x}|\boldsymbol{y})+\boldsymbol{\epsilon}\phi(t,\boldsymbol{\theta},\boldsymbol{x},\boldsymbol{c})+\nabla_{\boldsymbol{x}}\phi(t,\boldsymbol{\theta},\boldsymbol{x},\boldsymbol{c})\right)\right] \tag{27}$$

## B    Deferred Discussion

In this section, we continue our discussion from Sec. 4.3.

**How does baseline function reduce variance?**    The baseline function $\phi$ can be regarded as a guidance introduced into the distillation process. We contend that control variates when equipped with pre-trained models incorporating appropriate 2D/3D prior knowledge, are likely to exhibit a higher correlation with the score function. Intuitively, enforcing priors and constraints on the gradient space can also stabilize the training process by regularizing the optimization trajectory. Therefore, in our empirical design, the inclusion of geometric information expressed by a pre-trained MiDAS estimator is expected to result in superior variance reduction compared to SSD and VSD.

**Comparison with VSD.**    In VSD, the adopted control variate $\nabla\log q_t(\boldsymbol{x}|\boldsymbol{c})$ is fine-tuned based on a pre-trained score function using LoRA (Hu et al., 2021). However, this approach presents two primary drawbacks: 1) The trained control variate may not fully converge to the desired score function, potentially resulting in non-zero mean and biased gradient estimations. 2) Fine-tuning another large diffusion model also significantly increases the computation expenses. Our SSD effectively circumvents these two limitations. Firstly, the control variate in SSD is provably zero-mean, as per Stein's identity. Additionally, the computational cost associated with differentiating the frozen $\phi$ and optimizing the weights $\boldsymbol{u}$ remains manageable. We verify computational efficiency of SSD in Appendix C.2.

## C    Additional Experiments

### C.1    Quantitative Results

In addition to the qualitative comparison in Fig. 3, we also provide a numerical evaluation of these results in Tab. 1. Our observations indicate that SteinDreamer consistently outperforms all other methods, which improves CLIP score by ˜0.5 over ProlificDreamer. This superior result suggests our flexible control variates are more effective than the one adopted by ProlificDreamer.

### C.2    Wall-Clock Time Benchmarking

In addition to faster convergence, we also test per-iteration wall-clock time for all methods. Results are listed in Tab. 2. The reported numbers are obtained by averaging the running time of corresponding methods with six prompts in Tab. 1 for 10k iterations *on the same device*. In summary, SteinDreamer exhibits comparable per-iteration speed to DreamFusion while significantly outperforming ProlificDreamer in terms of speed. The trainable component $\boldsymbol{\mu}$ in SSD comprises only thousands of parameters, which minimally increases computational overhead and becomes much more efficient than tuning a LoRA in VSD. Notably, given that SSD can reach comparable visual quality in fewer steps, SteinDreamer achieves significant time savings for 3D score distillation.

| Methods | "blue tulip" | "sushi car" | "lionfish" |
|---|---|---|---|
| Dreamfusion (Poole et al., 2022) | 0.777 | 0.862 | 0.751 |
| ProlificDreamer (Wang et al., 2023b) | 0.751 | 0.835 | 0.749 |
| SteinDreamer w/ Depth (Ours) | **0.734** | **0.754** | **0.735** |
| | "cheesecake castle" | "dragon toy" | "dog statue" |
| Dreamfusion (Poole et al., 2022) | 0.902 | 0.904 | 0.789 |
| ProlificDreamer (Wang et al., 2023b) | 0.843 | 0.852 | 0.775 |
| SteinDreamer w/ Normal (Ours) | **0.794** | **0.806** | **0.751** |

Table 1: **Quatitative results.** We compare the CLIP distance ($\downarrow$ the lower the better) of demonstrated results among different approaches. Best results are marked in **bold** font. Prompts: "blue tulip" is short for "a blue tulip", "sushi car" for "a car made out of sushi", "lionfish" for "a lionfish", "cheesecake castle" for "a Matte painting of a castle made of cheesecake surrounded by a moat made of ice cream", "dragon toy" for "a plush dragon toy", and "dog statue" for "michelangelo style statue of dog reading news on a cellphone".

| Methods | Sec. / Iter. |
|---|---|
| Dreamfusion (Poole et al., 2022) | $1.063 \pm 0.002$ |
| ProlificDreamer (Wang et al., 2023b) | $1.550 \pm 0.004$ |
| SteinDreamer w/ Depth (Ours) | $1.093 \pm 0.005$ |
| SteinDreamer w/ Normal (Ours) | $1.087 \pm 0.004$ |

Table 2: **Benchmarking Wall-clock time**. We test wall-clock time (seconds per iteration) for all considered methods.

## C.3 LONGER TRAINING FOR BASELINES

A naive solution to achieve better convergence with high-variance gradients is to increase training steps. We test this hypothesis in this section by training DreamFusion and ProlificDreamer on two scenes with 10k more steps. Qualitative results are presented in Fig. 8. We notice that longer training time cannot guarantee better convergence. We also quantitatively find that more optimization steps have negligible influence on the final CLIP scores, which float between 0.84 ~0.86 for the prompt " car made out of sush" and 0.74 ~0.75 for the prompt "a lionfish".

In optimization theory, variance plays a crucial role in determining the convergence rate of SGD algorithms (Garrigos & Gower, 2023) With a finite number of optimization steps and a standard learning rate, maintaining low variance is pivotal to ensure convergence. Training with noisy gradients introduces high instability, potentially resulting in a suboptimal solution or even divergence, as illustrated in Fig. 4a.

## C.4 MORE QUALITATIVE RESULTS

We demonstrate more results on scene generation in Fig. 9. Consistent with our observation in Sec. 5, our method yields smooth and consistent renderings. Video demos can be found in the supplementary materials.

DreamFusion                                    ProlificDreamer

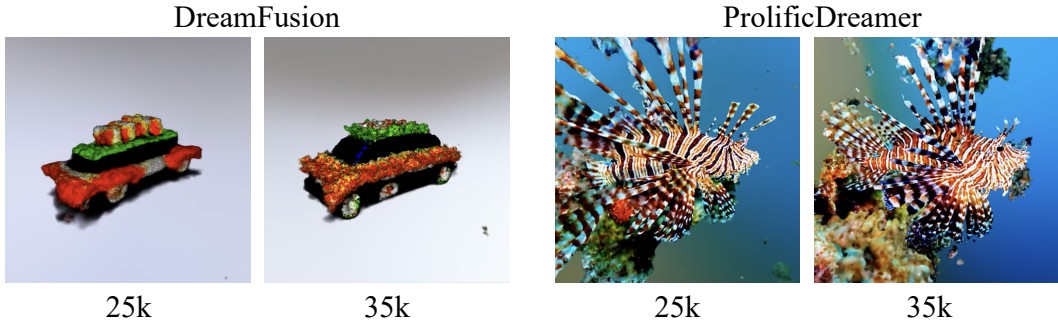

25k              35k                      25k              35k

Figure 8: **Longer Training Results.** We train high-variance score distillation approaches DreamFusion and ProlificDreamer for extra 10k steps. Prompts: " car made out of sush" for DreamFusion and "a lionfish" for ProlificDreamer

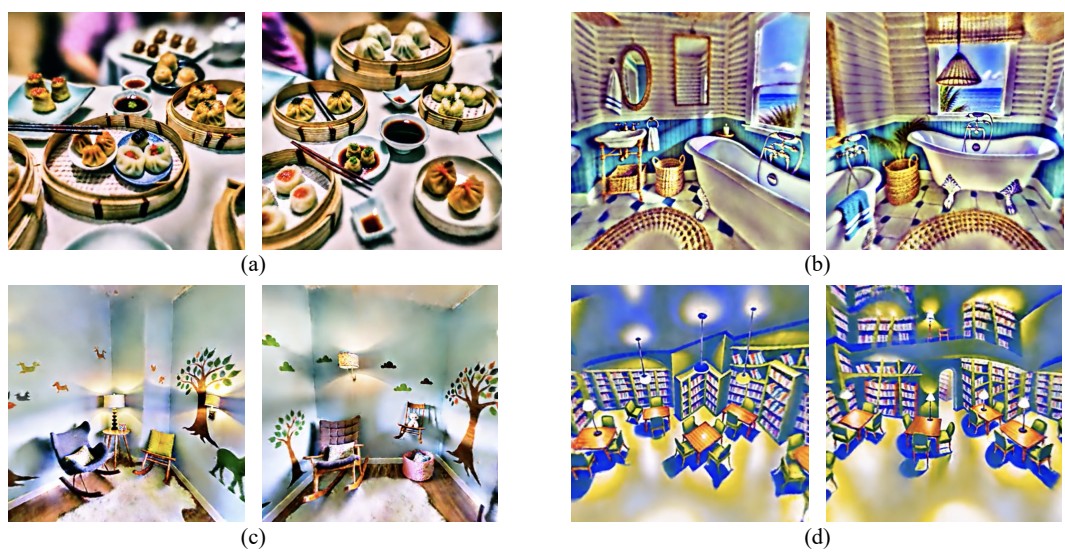

Figure 9: **More qualitative results by SteinDreamer.** Prompts: (a) "A DSLR photo of a table with dim sum on it", (b) "Editorial Style Photo, Eye Level, Coastal Bathroom, Clawfoot Tub, Seashell, Wicker, Blue and White", (c) "Editorial Style Photo, Wide Shot, Modern Nursery, Table Lamp, Rocking Chair, Tree Wall Decal, Wood", (d) "A library with tall bookshelves, tables, chairs, and reading lamps".

