# OpenReview forum: "SteinDreamer: Variance Reduction for Text-to-3D Score Distillation via Stein Identity"
_ICLR.cc/2024/Conference — Submitted to ICLR 2024_

### Official Review · Reviewer_4Z6k · 2023-10-29

**Soundness:** 3 good
**Presentation:** 3 good
**Contribution:** 3 good
**Rating:** 6
**Confidence:** 4

**Summary:**

This work proposes a more general solution to reduce variance for score distillation, termed Stein Score Distillation (SSD). SSD incorporates control variates constructed by Stein identity, allowing for arbitrary baseline functions. The experiment results demonstrate the effectiveness of the proposed method.

**Strengths:**

1. The paper is well-written.
2. The experiments show that the results are better than the baselines.
3. This work proposes to rethink the SDS/VSD in the way of variance, which is interesting.
4. The idea is novel.

**Weaknesses:**

1. My main concern is that why a lower variance during optimization is helpful for the final quality? Although there are some empirical results in Sec 3 show that there are some corelation between variance and generated quality, I think a more convincing justification should be given. Maybe a more detailed theoretical or intuitive explanation should be given.
2. No quantitative results are given to compare the proposed method and baselines in terms of the visual quality.

**Questions:**

1.  How will the baseline function effects the final results? Since the Stein identity always holds, what is the relationship between the baseline function and the final performance?
2. Can you show a 2D experiment? (Using SSD to directly optimize an image.) This will strengthen the effectiveness of SSD.

---

> ### Author Response · Authors · 2023-11-23
> **Response to Reviewer 4Z6k**
>
> We thank Reviewer 4Z6k for appreciating our theoretical reevaluation of SDS and VSD. Regarding your questions, please see our detailed responses below:
>
> **1. Why a lower variance during optimization is helpful for the final quality?**
>
> In optimization theory, variance plays a crucial role in determining the convergence rate of SGD algorithms (refer to Theorem 5.3 in [1]). With a finite number of optimization steps and a standard learning rate, maintaining low variance is pivotal to ensure convergence. Training with noisy gradients introduces high instability, potentially resulting in suboptimal solution or even divergence, as illustrated in Fig. 4(a).
>
> [1] Garrigos & Gower, Handbook of Convergence Theorems for (Stochastic) Gradient Methods
>
> **2. Quantitative results.**
>
> We present a qualitative comparison among DreamFusion, ProlificDreamer, and SteinDreamer across all showcased scenes, utilizing the CLIP score as our metric.
>
> | method              |   tulip |   sushi car |   lionfish |
> |:--------------------|--------:|------------:|-----------:|
> | DreamFusion      |   0.777 |       0.862 |      0.751 |
> | ProlificDreamer     |   0.751 |       0.835 |      0.749 |
> | SteinDreamer  |   0.734 |       0.754 |      0.735 |
>
> | method              |   cheesecake castle |   dragon toy |   michelangelo dog |
> |:--------------------|--------------------:|-------------:|-------------------:|
> | DreamFusion      |               0.902 |        0.904 |              0.789 |
> | ProlificDreamer     |               0.843 |        0.852 |              0.775 |
> | SteinDreamer |               0.794 |        0.806 |              0.769 |
>
> Our observations indicate that SteinDreamer consistently outperforms all other methods, which improves CLIP score by ~0.5 over ProlificDreamer. This superior result suggests our flexible control variates are more effective than the one adopted by ProlificDreamer.
>
>
> **3. The relationship between the baseline function and the final performance.**
>
> Various baseline functions can yield differing correlations between the constructed control variate and the target random variable for estimation. As outlined in Section 2.2, it is advisable to employ a baseline function that exhibits a stronger correlation with the 2D diffusion model to minimize variance. Typically, we notice that baseline functions with a more robust prior tend to demonstrate higher correlations. From another perspective, the baseline function $\phi$ can be regarded as a guidance introduced into the distillation process. Intuitively, enforcing priors and constraints on the gradient space can also stabilize the training process by regularizing the optimization trajectory. In our empirical design, the inclusion of geometric information expressed by a pre-trained MiDAS estimator is expected to result in superior variance reduction compared to SSD and VSD.

---

### Official Review · Reviewer_XeQE · 2023-10-31

**Soundness:** 3 good
**Presentation:** 2 fair
**Contribution:** 3 good
**Rating:** 6
**Confidence:** 4

**Summary:**

This work propose a novel regularization term to help SDS achieve low variance, named Stein Score Distillation (SSD). The proposed SSD is based on Stein's identity, which is natually zero-means to serve as variance control. The method starts from the insights that lower variance produces better performance with emprical results from DreamFusion and ProfilicDreamer. The experimental results demonstrate better shape, texture, and details on object and scene level generation.

**Strengths:**

- originality: the proposed method for text-to-3d generation is relatively novel, with inspiration from Stein's identity to reduce the variance in  the training process of text-to-3d.
- quality: this work starts from the emprical insights and combine with the proper mathematical solution to promote text-to-3d application. The experimental results outperform the previous reprsentative works, such as DreamFusion and ProfilicDreamer.
- clarity: the presentation of this work is good, with clear formluation and structure organization. It is reader friendly.
- significance: this work is of significance, especially in an age of AIGC.

**Weaknesses:**

- The experimental results are not as extensive as ProfilicDreamer. Compared with 10 object and 8 sence level generated content on https://ml.cs.tsinghua.edu.cn/prolificdreamer/, there are only 6 object and 4 sence level in this work and supplementary demo.
- Looking at the Figure 5, there existing the following confusions: 1) DreamFusion seems not converge for some cases, especially for "A Car made out of Sushi." and "A Lion Fish". Maybe DreamFusion needs more iterations? 2) The proposed SteinDreamer can be overfitting after ~120k training steps, with gradually increasing variance. How could this happen?
- Also, for comparing the convergence speed, it is much better to compare when all the methods are fully converged, especially for DreamFusion and ProfilicDreamer.
- Possible typos: 4th para. in Inroduction, 'aligns with with that' --> 'aligns with that'

**Questions:**

Please refer to the weaknesses.

---

> ### Author Response · Authors · 2023-11-23
> **Response to Reviewer XeQE**
>
> We appreciate Reviewer XeQE for recognizing our motivation and the clarity of our mathematical exposition. We have fixed all the typos and addressed your concerns as follows.
>
> **1. More experiment results.**
>
> We additionally showcase four qualitative results on scene generation in Appendix C.4. Aligning with our prior empirical observations, our method consistently produces sharp and visually coherent results.
>
> **2. DreamFusion results seem not converged.**
>
> Every qualitative result, including DreamFusion's, was obtained after 25k training steps, which follows the standard training configuration in the threestudio repository. To be more convincing, we train DreamFusion with the prompt "a car made out of sush" for an additional 10k steps, there were no observable improvements in the results. See our Appendix C.3 for more details.
>
>
> **3. Why does the variance of SteinDreamer gradually increase after 12k steps.**
>
> As depicted in Fig. 5, not all scenarios witness an increase in variance, as seen in examples like 'a blue tulip' and 'a plush dragon toy'. We emphasize that the optimization trajectory of score distillation exhibits high non-smoothness, leading to a typical fluctuation phenomenon in variance. Even in instances where variance appears to rise, its final value remains upper bounded, ensuring the convergence of our results. In contrast, other methods, such as ProlificDreamer with the prompt 'a plushy dragon toy', display potential variance explosion, evident in the training instability depicted in Fig. 4(a).
>
> **4. Compare convergence after more training steps.**
>
> We express our apologies for the mistake in Fig. 7, where the maximum of x-axis was incorrectly set to 100 instead of the intended 25k steps. This initial error occurred as we mistakenly used the indices of data points for the x-axis. We have rectified this issue in our revised version. It is important to note that 25k is the default setting suggested by threestudio, typically resulting in converged results with ProlificDreamer. To verify this claim, we train ProlificDreamer for an additional 10k steps. Both visual quality and the CLIP score exhibit negligible changes. See Appendix C.3 for more details.

---

### Official Review · Reviewer_D8i1 · 2023-11-01

**Soundness:** 3 good
**Presentation:** 2 fair
**Contribution:** 3 good
**Rating:** 5
**Confidence:** 3

**Summary:**

This paper proposes a variance reduction method for training text-to-3d synthesis models based on pretrained diffusion models. The authors first point out that prior work in this domain share the the same gradient in expectation, but differs in the Monte Carlo estimator. Then, they show that the performance difference can be explained by the variance of the estimator. Given this observation, they proposes a new variance reduction technique based on Stein's identity that generalizes the estimator in a previous work (SSD). The new estimator is shown to have lower variance and improve the visual quality of 3d scenes in experiments.

**Strengths:**

* The proposed method is well-motivated. Given the observation that prior gradient estimators have the same expectation and the performance is highly dependent on variance, it is very sensible to investigate better variance reduction techniques for this approach.

* The method is shown to have less variance than prior methods in experiments.

* The generated 3d scenes are visually better than prior methods (e.g., in Figure 3).

**Weaknesses:**

* Introducing another network for variance reduction increases the training cost per iteration, which might outweigh the benefit brought by having lower variance per iteration. It would be more convincing to include a wall-clock time comparison between different methods.

* One biggest weakness of this work is the insufficient empirical evaluation. The variance plot is one face of the story. However, it would greatly strengthen the work if the authors can show the 3d synthesis model learned by SSD is quantitatively better than the baselines given the same compute. In Figure 7, the improvement is very marginal considering the additional cost of training the control variates.

* The work can be positioned better in the literature. Sticking the landing  is not cited. The citations on Stein's identity can be improved. The generalized form of the Stein's identity in (7) is first introduced in Gorham & Mackey (2015).


Ref:
[1] Roeder, G., Wu, Y., & Duvenaud, D. K. (2017). Sticking the landing: Simple, lower-variance gradient estimators for variational inference. Advances in Neural Information Processing Systems, 30.

[2] Gorham, Jackson, and Lester Mackey. "Measuring sample quality with Stein's method." Advances in neural information processing systems 28 (2015).

**Questions:**

I would potentially raise the score if the wall-clock time comparison is included.

---

> ### Author Response · Authors · 2023-11-23
> **Response to Reviewer D8i1**
>
> We are grateful to Reviewer D8i1 for recognizing our motivation. We have conducted benchmarking of the wall-clock time and provided detailed per-point answers below.
>
> **1. Wall-clock time comparison between different methods.**
>
> Below, we list the average per-iteration wall-clock time for all methods, each benchmarked using the same GPU device with 10k training steps.
>
> | Method   | Per-iteration Time (s)  |
> | -------- | ------- |
> | DreamFusion  | 1.063 $\pm$ 0.002    |
> | ProlificDreamer | 1.550 $\pm$ 0.004     |
> | SteinDreamer    | 1.093 $\pm$ 0.005    |
>
>
> In summary, SteinDreamer exhibits comparable per-iteration speed to DreamFusion while significantly outperforming ProlificDreamer in terms of speed. The trainable component $\mu$ in SSD comprises only thousands of parameters, which minimally increases computational overhead and becomes much more efficient than tuning a LoRA in VSD. Notably, given that SSD can reach comparable visual quality in fewer steps, SteinDreamer achieves significant time savings for 3D score distillation.
>
> **2. Quantitative comparison with other methods.**
>
> We present a qualitative comparison among DreamFusion, ProlificDreamer, and SteinDreamer across all showcased scenes, using the CLIP score as our metric.
>
> | method              |   tulip |   sushi car |   lionfish |
> |:--------------------|--------:|------------:|-----------:|
> | DreamFusion      |   0.777 |       0.862 |      0.751 |
> | ProlificDreamer     |   0.751 |       0.835 |      0.749 |
> | SteinDreamer  |   0.734 |       0.754 |      0.735 |
>
> | method              |   cheesecake castle |   dragon toy |   michelangelo dog |
> |:--------------------|--------------------:|-------------:|-------------------:|
> | DreamFusion      |               0.902 |        0.904 |              0.789 |
> | ProlificDreamer     |               0.843 |        0.852 |              0.775 |
> | SteinDreamer |               0.794 |        0.806 |              0.769 |
>
> Our observations indicate that SteinDreamer consistently outperforms all other methods, demonstrating an approximate 0.5 improvement in CLIP distance over ProlificDreamer.
>
> **3. Missing citations.**
>
> Thank you for highlighting these two works. We have updated our citation to include mention of these references.
>
> [1] Roeder et al. Sticking the landing: Simple, lower-variance gradient estimators for variational inference. NeurIPS 2017.
>
> [2] Gorham et al. Measuring sample quality with Stein's method. NeurIPS, 2015.

---

### Official Review · Reviewer_HqJT · 2023-11-07

**Soundness:** 2 fair
**Presentation:** 2 fair
**Contribution:** 1 poor
**Rating:** 3
**Confidence:** 5

**Summary:**

This paper introduces an interpretation of score distillation sampling by integrating the Stein identity. This interpretation underscores the significance of selecting appropriate control variates for reducing variance to achieve efficient convergence during training. The authors propose the use of a monocular depth estimator for efficiency and demonstrate that their method effectively reduces variance during updates, leading to faster convergence compared to other score distillation sampling techniques.

**Strengths:**

- The paper is well-written and easy to understand.
-The adoption of a monocular depth estimator is a simple yet effective approach to reducing variance in text-to-3D synthesis.

**Weaknesses:**

**Limited novelty**
- It is noted that the primary focus of this paper is to interpret score distillation sampling using the Stein identity and to reduce variance through the incorporation of monocular depth estimation. However, Kim et al [1] have already discussed a similar interpretation of the Stein identity and the importance of baseline function selection for convergence and efficiency. Although the specific focus of Kim et al. is different (visual editing), the core inspiration appears to be similar. Therefore, it is recommended for the authors to acknowledge and discuss the already incorporated findings of Kim et al. In this context, while the inclusion of monocular depth estimation for efficient text-to-3D synthesis is intriguing, the novelty of the proposed method may be considered weak. Thus, given the similarities in findings and interpretation, the distinguishing factor appears to be only the choice of baseline functions for specific tasks, the adoption of a monocular depth estimator as the control variate.

**Lack of Quantitative Evaluation**
- To support their claims of qualitative improvement over Score Distillation Sampling (SDS) and Variational Score Distillation (VSD), the authors should provide a quantitative evaluation. Although the video demonstration provided by the authors indicates some improvement, issues such as the "Janus problem" in the dog statue are still present. While Figure 5 does demonstrate a reduction in variance compared to other methods, it does not necessarily guarantee improved quality. Figure 5 suggests that SSD reduces variance during training when compared to other methods, which may lead to fewer artifacts at similar training iterations, especially in comparison to ProlificDreamer. However, quality improvements are not guaranteed after fully training ProlificDreamer until convergence, as fine-tuning the diffusion model may require more computational time. Hence, it is recommended that the authors claim that the proposed method effectively reduces variance during training, which ensures better quality at an “early stage” not overall quality improvements. This concept is also supported by Figure 7 (although additional figures may be necessary for validation, just one example is severely not enough), clearly indicating that the reduced variance achieved by the proposed method results in faster convergence rather than overall quality improvement.
- Additionally, the comparison in the paper appears to be based on a limited number of samples. It is beneficial to conduct experiments with a broader range of prompts to assess the robustness of the proposed method in reducing variance.

[1] Kim et al., Collaborative Score Distillation for Consistent Visual Editing. NeurIPS 2023.

**Questions:**

As demonstrated in the Weaknesses section, the authors should make a comparison with the prior work of Kim et al. and discuss any differences if there exists new interpretation.

---

> ### Author Response · Authors · 2023-11-23
> **Response to Reviewer HqJT**
>
> We thank Reviewer HqJT for the time and effort in reviewing our paper. Per your questions, please see our response below:
>
> **1. Lack of novelty and idea coincides Kim et al.**
>
> Before submission, we were aware of Kim et al.'s work [1]. However, we respectfully disagree regarding the similarity of their method to ours. While both methods are grounded in Stein's method, the underlying principles significantly differ. In their approach [1], the SVGD-based update takes the form of the Stein discrepancy: $\max_{\phi \in \mathcal{F}} \mathbb{E}\_{x\sim q(x)} [\phi(x) \nabla \log p(x) + \nabla \phi(x)]$, where $\phi(x)$ is often interpreted as an update direction constrained by a function class $\mathcal{F}$ (RBF RKHS in [1]). In contrast, our update rule appends a zero-mean random variable via the Stein identity after the raw gradient of the KL divergence: $\mathbb{E}_{x \sim q(x)} [\nabla \log p(x) + \phi(x) \nabla \log q(x) + \nabla \phi(x)]$, where $\phi(x)$ typically represents a pre-defined baseline function. The potential rationale behind [1]'s approach to reducing variance lies in introducing the RBF kernel as a prior to constrain the solution space by modeling pairwise relations between data samples. Our approach to reducing variance is centered around constructing a more general control variate that correlates with the random variable of interest, featuring zero mean but variance reduction. We have cited [1] and included the above discussion in our revised submission (Sec. 4.3).
>
> [1] Kim et al., Collaborative Score Distillation for Consistent Visual Editing. NeurIPS 2023.
>
>
> **2. Clarification on training steps and quantitative results.**
>
> The qualitative results presented in this study were generated using a default setting of 25k training steps, consistent with the threestudio repo's default configuration. An error was identified in the original Fig. 7, where the x-axis required scaling to 25k due to our oversight in using the number of data points as the x-axis reference. We have rectified this error in the updated Fig. 7. To verify 25k is sufficient for convergence, we extended the training of ProlificDreamer by an additional 10k steps. During this extension, the CLIP score ranged between 0.74 and 0.75. Furthermore, a quantitative comparison is provided below.
>
>
> | method              |   tulip |   sushi car |   lionfish |
> |:--------------------|--------:|------------:|-----------:|
> | DreamFusion      |   0.777 |       0.862 |      0.751 |
> | ProlificDreamer     |   0.751 |       0.835 |      0.749 |
> | SteinDreamer  |   0.734 |       0.754 |      0.735 |
>
> | method              |   cheesecake castle |   dragon toy |   michelangelo dog |
> |:--------------------|--------------------:|-------------:|-------------------:|
> | DreamFusion      |               0.902 |        0.904 |              0.789 |
> | ProlificDreamer     |               0.843 |        0.852 |              0.775 |
> | SteinDreamer |               0.794 |        0.806 |              0.769 |
>
> The notably superior performance suggests that our method not only enhances quality at the early stages but also improves overall quality with full training.
>
> **3. Correction on Janus results.**
>
> We apologize for the inclusion of an incorrect video demo in the previous supplementary material, an oversight that occurred during the manuscript preparation rush. The updated supplementary material now accurately demonstrates that our method can potentially mitigate the Janus problem, although it is not the primary goal of this paper.
>
> **4. The number of examples is limited.**
>
> To enhance our experiment results, we present an additional set of four qualitative results on scene generation in Appendix C.4. As previously noted, our method consistently delivers smooth and visually consistent outcomes. Their video demos can be found in our updated supplementary materials.

---

### Meta-Review · Area_Chair_GATC · 2023-12-05

**Metareview:**

The paper introduces a novel method for text-to-3D generation, utilizing Stein's identity to reduce variance in the training process. The method shows promise in terms of reduced variance and improved visual quality of generated 3D scenes. However, as all of reviewers have mentioned, the paper lacks experimental evaluation. In particular, two reviewers suggested a quantitative evaluation, but the evaluation was made on only 6 3D objects. Due to the diversity of generative models, the provided empirical evidence is insufficient to convincingly demonstrate the superiority of the proposed method over existing methods like DreamFusion and ProlificDreamer. Given the significant concerns raised, the paper does not currently meet the high standards of ICLR. I recommend rejection. However, the authors are encouraged to address the highlighted issues and consider resubmission in future.

**Justification For Why Not Higher Score:**

The paper falls short in providing a comprehensive quantitative evaluation to substantiate its claims of improvement over existing methods such as DreamFusion or ProlificDreamer. The reviewers pointed out that the provided empirical evidence, mainly qualitative in nature, is not sufficient to convincingly demonstrate the superiority of the proposed method. The lack of extensive experiments weakens the paper's standing.

**Justification For Why Not Lower Score:**

N/A

---

### Decision · Program_Chairs · 2024-01-16

Reject